# Copy Number Variants in Two Northernmost Cattle Breeds Are Related to Their Adaptive Phenotypes

**DOI:** 10.3390/genes13091595

**Published:** 2022-09-06

**Authors:** Laura Buggiotti, Nikolay S. Yudin, Denis M. Larkin

**Affiliations:** 1Royal Veterinary College, University of London, London NW1 0TU, UK; 2The Federal State Budgetary Institution of Science Federal Research Center Institute of Cytology and Genetics, Siberian Branch of the Russian Academy of Sciences (ICG SB RAS), Novosibirsk 630090, Russia; 3Kurchatov Genomics Center, the Federal Research Center Institute of Cytology and Genetics, Siberian Branch of the Russian Academy of Science (ICG SB RAS), Novosibirsk 630090, Russia

**Keywords:** CNV, cattle, cold adaptation

## Abstract

Copy number variations (CNVs) are genomic structural variants with potential functional and evolutionary effects on phenotypes. In this study, we report the identification and characterization of CNVs from the whole-genome resequencing data of two northernmost cattle breeds from Russia: the Yakut and Kholmogory cattle and their phylogenetically most related breeds, Hanwoo and Holstein, respectively. Comparisons of the CNV regions (CNVRs) among the breeds led to the identification of breed-specific CNVRs shared by cold-adapted Kholmogory and Yakut cattle. An investigation of their overlap with genes, regulatory domains, conserved non-coding elements (CNEs), enhancers, and quantitative trait loci (QTLs) was performed to further explore breed-specific biology and adaptations. We found CNVRs enriched for gene ontology terms related to adaptation to environments in both the Kholmogory and Yakut breeds and related to thermoregulation specifically in Yakut cattle. Interestingly, the latter has also been supported when exploring the enrichment of breed-specific CNVRs in the regulatory domains and enhancers, CNEs, and QTLs implying the potential contribution of CNVR to the Yakut and Kholmogory cattle breeds’ adaptation to a harsh environment.

## 1. Introduction

Copy number variations (CNV) refer to a structural variation type where DNA segments of >1 kb are present in individual genomes in varying copy numbers, compared to a reference genome [1]. CNVs are less frequent than single nucleotide polymorphisms (SNPs) and other variations. However, they can potentially have a larger functional and evolutionary impact, such as changing gene structure and dosage, altering gene regulation, and exposing recessive alleles [2,3]. CNVs and their impacts have been extensively studied, especially in humans [4], where CNVs are considered to affect gene expression and, therefore, some phenotypes of interest. For example, CNV loss in the NPY4R gene was associated with obesity [5]. Other studies have revealed that genomic diversity could be increased due to the differential selection of CNVs for adaptations to different environments [6,7,8]. For instance, 30% of young fast evolving duplicated genes in sticklebacks are in CNVs and these genes are enriched in functional categories related to environmental adaptations [8]. Studies in livestock also highlight the role of CNVs in shaping various phenotypes. A partial or complete duplication of the *KIT* gene causes different patterns of white coat coloration in pigs and in some of the cattle breeds [9,10], while a white coat color in sheep has been associated with a duplication of the *ASIP* gene [11].

The present study focuses on CNV detection from the whole-genome resequencing data of two cold-adapted cattle breeds from Russia: the Kholmogory and Yakut cattle. Both breeds live in harsh environments but have very different population histories. The Kholmogory was formed in the European part of Russia around 300 years ago [12], while the Yakut cattle was formed at the Baikal area of Siberia and likely migrated together with the Yakut people to contemporary Yakutia about 500–800 years ago [13]. To identify CNVs and CNV regions (CNVRs) which could co-evolve with the adaptation of the Russian cattle breeds to harsh climate conditions, we utilized sequences of four breeds (Yakut, Kholmogory, Holstein, and Hanwoo). Previously, the Yakut cattle has been found to be related to Korean Hanwoo, and Kholmogory to Holstein [14]. Therefore, we compared Kholmogory and Yakut cattle to phylogenetically close breeds to search for CNVs that could have an influence on breed-specific biology and adaptations.

## 2. Materials and Methods

The Yakut and Kholmogory cattle breeds were previously whole-genome resequenced [15] and mapped to the reference Hereford cattle assembly (UMD3.1, BosTau6) using BWA-MEM [16] with default parameters; Hanwoo and Holstein resequencing data were downloaded from the Sequence Repository Archive [17]. A total of 98 high-quality samples of the four cattle breeds (Yakut (29), Kholmogory (32), Hanwoo (19), and Holstein (18)) were used. The cn.MOPS R package (copy number estimation by a Mixture of Poissons [18]) was used for CNV detection. Based on the average sequence coverage of our data (~11X), window length was set to 700 (windowLength = readLength × 50/coverage) and posterior probabilities were estimated (posteriorProbs). The cn.MOPS tool represents a CNV detection pipeline that models the depths of coverage across multiple samples at each genomic position. Using a Bayesian approach, it decomposes read variants across samples into integer CNVs and noise using mixture components and Poisson distributions, respectively. The multiple samples approach increases statistical power and decreases computational burden and the FDR in CNV detection. CNVs were then used to construct a set of copy number variable regions (CNVRs) for Kholmogory, Yakut, Hanwoo, and Holstein breeds. The CNVRs were constructed by merging CNVs across samples of the same breed that exhibited at least 50% pairwise reciprocal overlap in their genomic coordinates; unique CNVRs per breed were those with less than 10% overlap with CNVRs in other breeds. BEDTools [19] and BEDOPS [20] tools were used to calculate CNVR overlaps. Genomic Regions Enrichment of Annotations Tool (GREAT), [21] was used to assign each gene of the reference Hereford cattle assembly (UMD3.1, BosTau6) to a regulatory domain consisting of a basal domain that extends 5 kb upstream and 1 kb downstream from its transcription start site (total length of the regions per gene is 6000 bp). The karyoploteR [22] package was used to plot the cattle chromosome map and to visualize the CNVRs locations for the four breeds.

## 3. Results and Discussion

A total of 860,380 autosomal CNVs were detected in the four-breed set, which were then merged into 71,549 CNVRs. Interestingly, the Yakut and Kholmogory breeds shared the largest fraction of CNVRs (138.41 Mb). The second largest shared fraction of Kholmogory CNVRs was with Holstein (61.05 Mbp) and Yakut CNVRs with Hanwoo (27.92 Mbp), confirming known breed relations. A total of 19,502 CNVRs (total length: 106.09 Mbp) were breed-specific for the Yakut, 2238 (18.61 Mbp) for the Kholmogory, 2535 (8.27 Mbp) for the Hanwoo and 1625 (4.95 Mbp) for the Holstein cattle (Figure 1).

To reveal a possible contribution of CNVRs to breed-specific biology and adaptations, we investigated 5522 genes found in the four breed-specific CNVRs, of which 2962, 1034, 862, and 664 genes were found in the Yakut, Hanwoo, Kholmogory, and Holstein CNVRs, respectively (Appendix A). A gene ontology (GO) enrichment analysis highlighted distinct pathways being enriched in these gene sets, with the second largest number (35) found in the Yakut cattle, among which we observed cognition, the regulation of small GTPase-mediated signal transduction, the detection of mechanical stimulus, etc. (Appendix A). The Kholmogory cattle had the third largest number of GO categories (22) of which intracellular signal transduction, GTPase regulator activity, adenyl nucleotide binding, etc. were uniquely present. The Hanwoo had the largest number of GO categories (38), while Holstein had the same number of GO categories as Kholmogory (22). Interestingly, all the breeds showed enrichment for the GO category response to stimulus, although the genes were different. Moreover, a DAVID functional annotation cluster analysis [23] highlighted the enrichment of ubiquitin protein in the Yakut cattle, which is involved in protein degradation and found to be enriched in Antarctic fish [24]. The authors hypothesized that the cost of living for cold-adapted ectotherms commits more effort to maintaining protein homeostasis. Moreover, ATPase activity, microtubule motor activity, and blood coagulation inhibitor were also found to be enriched in the Yakut cattle CNVRs, which could potentially influence thermoregulation (Appendix A). There were 7414 CNVRs shared between the Yakut and Kholmogory breeds, overlapping 2925 genes, which were enriched in various pathways such as keratin filament, kinase, oxytocin signaling pathway, etc. (Appendix A).

Yakut cattle-specific CNVRs involved multiple fatty-acid related genes such as the *CYP4A11* (cytochrome P-450 4A11), which is implicated in lipogenesis and growth traits. This CNVR has previously been reported in various Chinese native taurine cattle breeds (Jaxian, Quinchuan, Nanyang, Jinnan, Luxi, and Chinese Red Steppe [25]). The Kholmogory cattle unique CNVRs covered over one-thousand annotated genes, among which a few were interleukin genes (*IL17RE*, *IL20RA*, *IL10RA*, *IL10*) associated with immune response, as well as *PLA2G4A*, which is involved in inflammatory responses [26]. To further investigate the potential association of CNVRs with gene regulation we investigated the landscape of breed-specific CNVRs overlapping regulatory domains by using the GREAT approach [21]. We found 1109, 663, 659, and 516 genes with breed-specific CNVRs in their GREAT domains in Yakut, Holstein, Hanwoo, and Kholmogory cattle breeds, respectively. The GO enrichment analysis of the latter sets of genes found the response to stimulus to be enriched in the Yakut but not in the Kholmogory cattle (Appendix A), while a DAVID functional annotation cluster analysis revealed the term lipid transport and lipoprotein metabolic process to be unique to Yakut cattle when considering only breed-specific CNVRs in GREAT domains (Appendix A), suggesting a potential fundamental role of CNVRs in regulating thermogenesis [27], a major adaptive mechanism to extreme climates. Moreover, the top DAVID functional cluster (Appendix A) contained genes that are involved in cytoskeletal reorganization (such as *PADI-* genes), which also plays a role in shaping the adaptation to a cold environment [28]. Kholmogory, on the other hand, had the unique GO terms: metal ion transport, cellular response to organic substance and transporter activity, among others, suggesting again their potential role in the adaptation to new ecological niches [29,30]. Finally, we investigated the enrichment of breed-specific CNVRs in three functional classes: (i) conserved non-coding elements (CNEs); (ii) enhancers; (iii) QTLs regions, by using GAT [31]. Interestingly, breed-specific CNVRs were enriched with CNEs [32] in all four breeds, highlighting again their potential involvement in gene regulation, and Yakut cattle-specific CNVRs were significantly negatively enriched of both cetartiodactyla and ruminant CNEs (Figure 2). The QTLs enrichment analysis suggests that Yakut cattle-specific CNVRs are significantly negatively enriched in all major QTLs categories but meat and carcass, while CNVs in the other three breeds show positive or negative associations with a limited number of QTL categories (Figure 2). One explanation for the QTLs enrichment results is that in a natural population subjected to a harsh climate such as Yakut cattle, selection acts on CNVRs contributing to phenotypes that are quite different from those which are normally under selection by humans.

## 4. Conclusions

Overall, our results point to novel copy number variants and their potential contributions to local adaptations in the northernmost cattle breeds from Russia and shared CNV events involving GO terms such as a response to stimulus related to thermoregulation. The enrichment of ubiquitin proteins in Yakut cattle unique CNVRs might indicate their contribution to maintaining protein homeostasis; moreover, the enrichment of multiple fatty-acid related genes implicated in lipogenesis and growth traits suggests their potential involvement in thermoregulation, as well as the petite size of the Yakut cattle breed.

Finally, both the Yakut and Kholmogory cattle breeds had breed-specific CNVRs enriched in regulatory domains and enhancers, CNEs, and QTLs highlighting their potential contribution to harsh environment adaptations.

## Figures and Tables

**Figure 1 genes-13-01595-f001:**
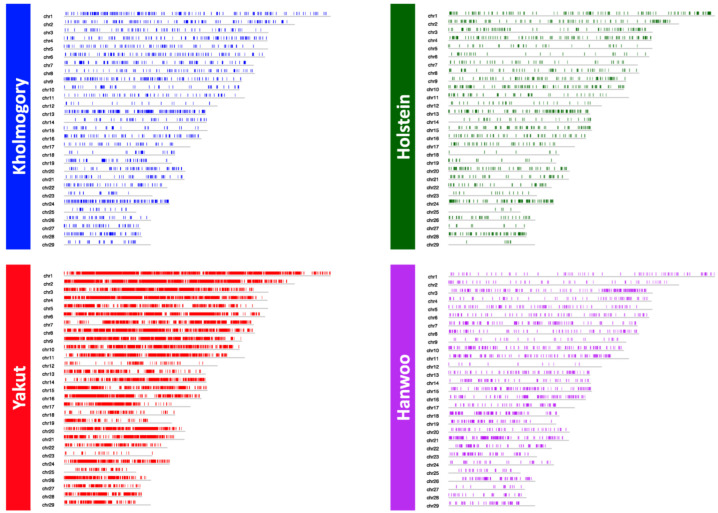
Whole genome distribution of breed-specific CNVRs for the four breeds.

**Figure 2 genes-13-01595-f002:**
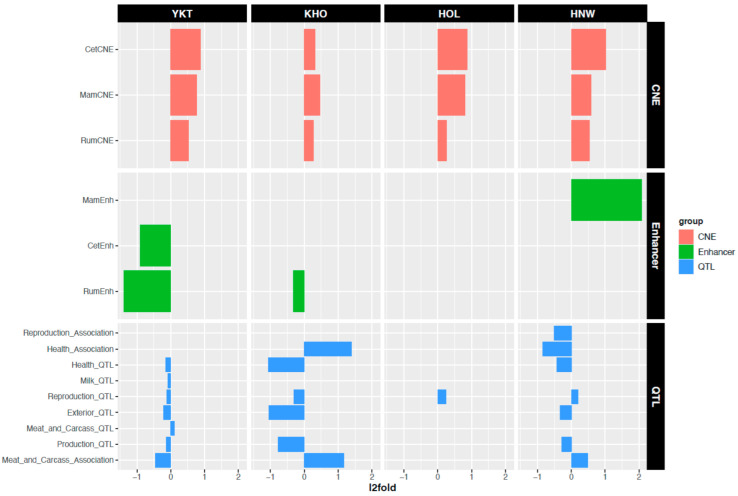
Breed-specific CNVR overlapping CNEs, enhancers, and QTLs. Abbreviations are as follows: YKT—Yakut, KHO—Kholmogory, HOL—Holstein, HNW—Hanwoo, CNE—conserved non-coding elements.

## Data Availability

Whole genome sequences data that support the findings of this study have been deposited in GenBank (PRJNA642008).

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
