# Peer review of "Copy Number Variants in Two Northernmost Cattle Breeds Are Related to Their Adaptive Phenotypes"

_genes, 2022, doi:10.3390/genes13091595_

Round 1

Reviewer 1 Report

Dear authors

The introduction is clearly written, yet some more examples about the importance of CNVs or any associations with adaptation or other phenotypic traits would be useful. Introduction is rather short and the manuscript could gain by extending the above mentioned subjects.

Materials and methods also lack of details (p.ex: Line 60: Please elaborate more about window length and probabilities). I would prefer a more analytical approach

Figure 1: the names of the breeds should be of just a white font without this shadow, to be clear to the reader. however this is a very nice and interesting figure. What program was used for this figure? is it from the R package?

Did you check for genes and GO enrichment in common CNVRs in all breeds (if there are any)?

Lines 112-114 belong to M&M to my opinion, and you should present more details about GREAT

Author Response

We are very grateful to Reviewer 1 for their time and effort to read our manuscript and for their comments. Our responses are included below:

Point 1: The introduction is clearly written, yet some more examples about the importance of CNVs or any associations with adaptation or other phenotypic traits would be useful. Introduction is rather short, and the manuscript could gain by extending the above mentioned subjects.

Response 1: We added one example on fast evolving genes in the stickleback CNVs related to environmental adaptations but feel that due to the manuscript’s short format we cannot extend on the introduction too much.

Point 2: Materials and methods also lack of details (p.ex: Line 60: Please elaborate more about window length and probabilities). I would prefer a more analytical approach.

Response 2: Thank you. We have further elaborated on the window length and posterior probabilities in Materials and Methods.

Point 3: Figure 1: the names of the breeds should be of just a white font without this shadow, to be clear to the reader. however this is a very nice and interesting figure. What program was used for this figure? is it from the R package?

Response 3: the shadows on the breed names were probably caused by a formatting issue, we have replaced the figure in the revised manuscript. The R package karyoploteR was used to make the figure and was added to the Materials and Methods section.

Point 4: Did you check for genes and GO enrichment in common CNVRs in all breeds (if there are any)?

Response 4: as the focus of this short communication was to identify CNVs and CNVRs which could co-evolve with adaptation of the cattle breeds to harsh and cold climate conditions we explored the Yakut and Kholmogory unique CNVRs only.   

Point 5: Lines 112-114 belong to M&M to my opinion, and you should present more details about GREAT

Response 5: details about GREAT have been added to the Materials and Methods section.   

Reviewer 2 Report

In this study, Laura et al. reported the identification of copy number variations from whole-genome sequencing data of two northernmost cattle breeds, they also found these copy number variations are related to thermoregulation. There are some concerns that need to be addressed:

  1. Some additional descriptions of how the analysis of whole genome sequencing data has been conducted.
  2. It would be better if the authors could add some sentences to discuss their findings in the conclusion part.
  3. It would be great if the authors could point out some thermoregulation genes that have a higher possibility of making difference in the adaptive phenotypes of different cattle breeds.

Author Response

We are very grateful to Reviewer 2 for their time and effort to read our manuscript and for their comments. Our responses are included below:

Point 1: Some additional descriptions of how the analysis of whole genome sequencing data has been conducted.

Response 1: please refer to “Buggiotti L, Yurchenko AA, Yudin NS, Vander Jagt CJ, Vorobieva NV, Kusliy MA, et al. Demographic history, adaptation, and NRAP convergent evolution at amino acid residue 100 in the world northernmost cattle from Siberia. Mol Biol Evol. 2021” for a full description of the analysis of the whole genome sequences. We referenced this paper in the Methods section for mapping of reads. 

Point 2: It would be better if the authors could add some sentences to discuss their findings in the conclusion part.

Response 2: additional sentence has been added to the conclusion section.

Point 3: It would be great if the authors could point out some thermoregulation genes that have a higher possibility of making difference in the adaptive phenotypes of different cattle breeds.

Response 3: Indeed, we found enrichments of breed-specific CNVRs in regulatory regions of genes and conserved non-coding elements related to thermoregulation genes. We now have highlighted some of these genes in the revised manuscript.